# The Role of Peritumoral Depapillation and Its Impact on Narrow-Band Imaging in Oral Tongue Squamous Cell Carcinoma

**DOI:** 10.3390/cancers15041196

**Published:** 2023-02-13

**Authors:** Andrea Iandelli, Claudio Sampieri, Filippo Marchi, Alessia Pennacchi, Andrea Luigi Camillo Carobbio, Paola Lovino Camerino, Marta Filauro, Giampiero Parrinello, Giorgio Peretti

**Affiliations:** 1IRCCS Ospedale Policlinico San Martino, 16132 Genoa, Italy; 2Department of Surgical Sciences and Integrated Diagnostics (DISC), University of Genova, 16126 Genoa, Italy; 3Section of Otorhinolaryngology—Head and Neck Surgery, Department of Neurosciences, University of Padua—“Azienda Ospedaliera di Padova”, 35128 Padua, Italy

**Keywords:** oral tongue squamous cell carcinoma, depapillation, perineural invasion, narrow-band imaging, depth of invasion, head and neck cancer

## Abstract

**Simple Summary:**

The present study evaluates the reliability of depapillation surrounding oral tongue squamous cell carcinomas (OTSCC) as a perineural invasion predictor and how it could affect narrow-band imaging performance. Our retrospective study was conducted on seventy-six patients affected by OTSCC submitted to radical surgery. The presence of depapillation, clinicopathological variables and narrow-band imaging vascular patterns were analyzed. Our results report that peritumoral depapillation is a reliable feature for perineural invasion in OTSCC, and narrow-band imaging margin detection is not impaired by depapillation.

**Abstract:**

A recent study reported that the occurrence of depapillated mucosa surrounding oral tongue squamous cell carcinomas (OTSCC) is associated with perineural invasion (PNI). The present study evaluates the reliability of depapillation as a PNI predictor and how it could affect narrow-band imaging (NBI) performance. This is thus a retrospective study on patients affected by OTSCC submitted to radical surgery. The preoperative endoscopy was evaluated to identify the presence of depapillation. Differences in distribution between depapillation and clinicopathological variables were analyzed. NBI vascular patterns were reported, and the impact of depapillation on those was studied. We enrolled seventy-six patients. After evaluation of the preoperative endoscopies, 40 (53%) patients had peritumoral depapillation, while 59 (78%) had a positive NBI pattern. Depapillation was strongly correlated to PNI, 54% vs. 28% (*p* = 0.022). Regarding the NBI pattern, there was no particular association with depapillation-associated tumors. The presence of depapillation did not affect the intralesional pattern detected by the NBI, while no NBI-positive pattern was found in the depapillation area. Finally, the NBI-guided resection margins were not affected by depapillation. Peritumoral depapillation is a reliable feature for PNI in OTSCC. NBI margin detection is not impaired by depapillation.

## 1. Introduction

Oral cavity squamous cell carcinoma (OSCC) represents a considerable burden of global neoplastic disease. OSCC accounts for more than fifty thousand new diagnoses per year in the United States [1], with a reported cancer-specific mortality of around 30% [2,3,4,5]. The tongue is affected in about half of all cases [6], demonstrating a tendency to local tissue and regional lymphatic spread early in the course of the disease.

Despite the progress made in multidisciplinary care, oncologic outcomes in patients affected by oral tongue squamous cell carcinoma (OTSCC) did not show significant changes over the last two decades, and surgery remains the mainstay of treatment [7,8]. Currently, the Union for International Cancer Control (UICC)/American Joint Committee on Cancer (AJCC) 8th edition TNM staging system for oral cancer identifies several pathologic risk factors that warrant adjuvant treatment [9,10]. 

Among those, perineural invasion (PNI) is known to be an independent prognostic factor for overall survival (OS), disease-specific survival (DSS), and locoregional recurrence [4,11]. Moreover, this histological feature correlates with the presence of lymph node metastasis, representing a strong prognosticator for poor outcomes, especially in early-stage tongue neoplasms [12,13,14,15]. Therefore, PNI could be considered a surrogate histological marker that defines a tumor’s aggressive behavior and could guide the surgeon in the therapeutic management and adjuvant approach [16]. 

Depapillation is a clinical feature that may be encountered on the tongue epithelium. It appears as a strip of mucosa that is smoother than the normal tongue surface due to the papilla dysfunction. Recently, the presence of an evident strip of depapillated mucosa around OTSCCs was described for both early- and advanced-stage tumors [17]. It is believed that the local neoplastic invasion and inflammation cause the tongue’s papilla dysfunction, leading to the clinical appearance of depapillation [17]. This dysfunction would lead to the accumulation of neuro-promoters and the presence of a favorable environment for the neural infiltration of neoplastic cells. In their work, Singh and colleagues reported that peritumoral depapillation is indeed strongly associated with PNI [17]. 

A useful tool in the preoperative characterization of cancers of the oral cavity is narrow-band imaging (NBI). Thanks to the ability of NBI to reveal the submucosal vascular abnormalities induced by a tumor, Takano et al. suggested a classification of oral lesions into four categories according to the shape and size of the typical neoplastic intrapapillary capillary loops (IPCL) [18]. This classification proved to be effective in the evaluation and clinical diagnosis of OSCC [19,20]. Moreover, NBI was proven to be an important tool in guiding the surgeon during tumor resection, leading to a significant in superficial positive margins [21,22]. Over the years, the main issue described in the literature of this endoscopic technique has been its value on different oral cavity subsites. According to Lin et al., where keratinized mucosa shows a greater thickness than 1300 μm, capillary loops are difficult to evaluate, such as in the dorsal tongue [23]. However, recently, Piazza et al. addressed this topic and described how epithelial thickness and keratinization do not hinder the penetration of blue and green wavelengths since the papillae of the lamina propria and IPCLs usually reach a more superficial layer, confirming NBI’s diagnostic value to be comparable in every oral cavity subsite [24]. In their work on 128 patients, a comparison of the diagnostic value of NBI did not show statistically significant differences among different oral cancer subsites characterized by three different types of epithelia: thick keratinized, thin non-keratinized, and thick non-keratinized. Furthermore, epithelial thickness is usually thought to be the distance from the epithelial surface to the basal membrane, but the papillae of the lamina propria with IPCLs reach a more superficial layer. Therefore, the actual depth of penetration needed to assess IPCLs is within the limit of NBI light penetration across all three different types of epithelia above-mentioned [24]. In the case of depapillation, a consideration needs to be addressed: potential changes in epithelium thickness in this scenario could prevent the identification of submucosal vascularization. The concern that depapillation could be linked to a change in epithelium thickness was described in a review by Picciani et al. in 2016 [25]. On the surface of the tongue mucosa, papillary epithelial atrophy could suggest parakeratosis and acanthosis underneath [25]. In literature, NBI’s efficacy in detecting the specific neoangiogenic patterns characteristic of premalignant and neoplastic transformation has been widely described [20,24,26,27,28,29]: well-demarcated brownish or darker areas in the context of a green–blue-appearing normal mucosa with thick dark spots, increased microvascular density, and the arrangement of vessels in intraepithelial papillary capillary loops (IPCLs) inside and surrounding the lesion. Additionally, if a lesion is characterized by a leukoplastic appearance, this typical microvascular arrangement may only be detected in the mucosa surrounding the cancer. Furthermore, the presence of afferent perpendicular thick vessels pointing towards the lesion and branching out in IPCLs within its context can be frequently observed. Moreover, neoplastic-induced neoangiogenesis leads to the elongation of IPCLs and the development of additional intraepithelial vessels that are not usually seen in normal mucosa. These superficial vascular abnormalities are easily detected by NBI light, even in oral subsites with thicker mucosa [24]. 

Our work aims to further investigate the relation between PNI and the presence of depapillation in OTSCC to potentially predict neural spreading during the preoperative clinical evaluation to help in the decision of the most appropriate therapeutic management for the patient. The secondary endpoint is to assess NBI’s diagnostic value in depapillated mucosa and evaluate if resection margin status guided by NBI is affected by depapillation.

## 2. Materials and Methods

### 2.1. Study Design

A single-center retrospective study was carried out at the Unit of Otolaryngology, Head and Neck Surgery, IRCCS Ospedale Policlinico San Martino, University of Genova (Genova, Italy), after approval by the Institutional Review Board that waived the need for informed consent due to the study’s retrospective nature. The study was carried out in accordance with the principles of the Helsinki Declaration. 

The primary objective was to determine whether the presence of depapillation was associated with other clinical or histopathological factors in oral tongue cancers. The secondary objective was to study the correlation between depapillation and NBI. All patients who underwent surgery with curative intent from September 2013 to March 2022 were reviewed. The inclusion criteria were as follows: (1) biopsy-proven oral cavity invasive squamous cell carcinoma; (2) the presence of a preoperative endoscopic recording with a white light (WL) and NBI examination; (3) the presence of a pathological report after definitive excision; (4) a primary lesion centered on or extending to the lateral border/dorsum of the mobile tongue. Exclusion criteria were: (1) previous surgeries in the oral cavity; (2) the previous administration of RT in the oral cavity; (3) the absence of tumor extension to the tongue papillae; (4) the presence of any comorbidity or associated potentially malignant disorder that could possibly influence the presence of papillary atrophy on the tongue.

### 2.2. Treatment Protocol

A preoperative panendoscopy was conducted in all cases in the office with a flexible digital endoscope through the nose and rigid telescope through the mouth with an Olympus CV-170 ENT digital platform together with a video-endoscope ENF-VT2 (Olympus Medical System Corporation, Tokyo, Japan). All the endoscopic examinations were carried out in WL and then switched to NBI to better define the local extension of the tumor and to find possible satellite lesions or second primaries. Primary tumor excision was planned based on the endoscopic extension and the preoperative imaging (CT scan and/or MRI). Distant metastases were ruled out preoperatively either with a PET CT or chest CT coupled with an abdomen ultrasound examination. All patients had been submitted to surgery after multidisciplinary team (MDT) discussion and preoperative counseling between head and neck surgeons and radiation and medical oncologists. Therapeutic neck dissection was performed simultaneously according to the presence of nodal metastasis at presentation or electively if the preoperative depth of invasion (DOI) measured at the imaging was ≥4 mm. If not already performed, elective neck dissection was carried on after the primary excision in case of a pathological DOI ≥ 4 mm. Adjuvant RT was started 4–6 weeks after surgery in the presence of adverse pathological features and/or pathological DOI of 4 mm or more.

### 2.3. Data Collection

The following data were collected: age, gender, smoking and alcohol status, tumor appearance (exophytic, plane, or ulcerated) at the preoperative endoscopy, clinical TNM staging, pathological TNM staging, surgical margin status, grading, DOI, presence of lymphovascular invasion (LVI) or perineural invasion (PNI), presence of tumor-infiltrating lymphocytes (TILs), tumor budding [30], the worst pattern of invasion (WPOI) [30,31] at the primary specimen, and the presence of positive nodes during the neck dissection. The presence of lymph node metastasis was defined based on the final pathology report for those patients who underwent neck dissection and on the preoperative imaging for those who did not. A definitive histopathological report was used to define the surgical margin status, labeled as positive if microscopic invasive carcinoma was found at the inked margin of the specimen and close if at a distance inferior to 5 mm. All tumors were staged according to the 8th edition of the American Joint Committee on Cancer (AJCC) staging manual [32]. The above-mentioned risk factors, such as PNI and LVI, were gathered by blind data collectors who did not know if a lesion was characterized by the presence of depapillation.

Depapillation was defined according to the International Agency for Research on Cancer as “atrophy or absence of papilla” on the tongue [33], and it was assessed on the oral endoscopy video recordings where the primary tumor interface with the tongue’s papillae was clearly visible in WL (Figure 1).

The NBI vascular pattern was assessed only by the senior surgeon according to the classification proposed by Takano et al. based on IPCL, which is reported here [18]. The type I pattern comprises normal IPCL running parallel to the surface of the mucosa and appearing as a wavy line. In the Type II pattern, the IPCL have a similar shape to type I, but their caliber is notably increased and dilated. The IPCL of Type III are tangled lines (due to the severe increase in length) running perpendicular to the surface of the mucosa and therefore appearing as dark brown dots from a distance. Type IV is characterized by a chaotic representation of the IPCL that appear as large vessels with no loops at the terminal branches due to the progression of carcinogenesis, which leads to the dilation and elongation of the loops and finally to their destruction. Type IIIa and IV IPCLs are considered to be strongly associated with carcinoma in situ and invasive carcinoma in the oral cavity. Takano’s classification is depicted in Figure 2. Both depapillation and NBI patterns were assessed by a senior head and neck surgeon with special experience in oral cavity surgery and NBI (Figure 3). To assess if the identification of the depapillation and NBI pattern were reproducible and reliable, the senior surgeon trained a young fellow to recognize them. The young fellow assessed those independently and was blinded to the senior surgeon’s evaluation. For the univariable analysis, only the senior’s assessment was considered.

### 2.4. Statistical Analysis 

Categorical variables were summarized as counts and percentages, while continuous variables were reported as medians and interquartile ranges (IQR: 25th and 75th). Frequencies and distribution for all demographic, clinical and pathological variables of the cohort were reported. The difference in the distribution of categorical variables was compared using the chi-squared test and Fisher’s exact test, as appropriate. Differences in continuous variables between groups were tested using the Mann–Whitney U test. The influence of depapillation on recurrence-free survival (RFS), defined as the time from surgery to death or tumor recurrence, was also considered a secondary endpoint. Calculations were made using the Kaplan–Meier method, and the survivals among the two groups were compared with the log-rank test. To assess the inter-rater agreement, the Cohen’s Kappa coefficient was computed based on the evaluations made by the senior and the young authors, and it was interpreted as follows: values 0–0.20 as no agreement, 0.21–0.40 as fair, 0.41–0.60 as moderate, 0.61–0.80 as substantial, and 0.81–1.00 as (almost) perfect agreement. Finally, for all tests, a value of *p* < 0.05 was considered to indicate statistical significance. Statistical analyses were performed using the R software for statistical computing (R version 4.0.1). Data were analyzed in November 2022.

## 3. Results

Seventy-six patients met the inclusion criteria and were included in the analysis. The whole cohort had a median age of 70.00 (IQR: 56.0–80.0), and the majority of patients were females (n = 40, 53%). The main features of the study population are summarized in Table 1. After the evaluation of the preoperative endoscopies by the senior specialist, 40 (53%) patients had depapillation at the WL examination, while 59 (78%) had a Takano’s Type III or IV pattern with NBI.

When measuring the inter-rater agreements, both the accordance between the physicians in assessing the presence of depapillation and rating the NBI pattern were substantial and greater than would be expected by chance (κ = 0.63, Z = 5.59, *p* < 0.001 and κ = 0.80, Z = 7.01, *p* < 0.001, respectively).

The presence of depapillation was strongly correlated to PNI, which was found in 54% of the cases in the depapillation group vs. 28% of the cases in the non-depapillated group (*p* = 0.022). No other factor differed significantly between the two groups (Table 2). Notably, no statistically significant difference was found in the number of cases with positive margins between the depapillated lesions group and the group of lesions that was not characterized by depapillation. Regarding the NBI pattern, according to Takano et al., there was no correlation with depapillation.

The median follow-up time was 13.0 months (IQR: 3.00–35.50). For the whole cohort, the 2-year and 5-year RFS (95% CI; number still at risk) were 67% (0.55–0.82; 24) and 59% (0.45–0.77; 7), respectively. Figure 4 shows the Kaplan–Meier curves when comparing the RFS of different groups based on the presence of depapillation or the NBI pattern: no statistical differences were found with the log-rank test for both comparisons (*p* = 0.91 and *p* = 0.59, respectively).

## 4. Discussion

Histological adverse pathological factors (APF) can reveal tumor aggressiveness, and they impact local and regional control [34,35]. Predicting, preoperatively, the biological behavior of OTSCC would allow the surgeon to tailor the tumor excision or the neck treatment. This would be valuable, especially in early-stage OTSCC, as, due to histopathological heterogeneity, there are important variations in the outcome even with identical clinical stages. In fact, although stage I/II OTSCC has been shown to have favorable outcomes, an estimated 30–35% of patients have locoregional recurrences [36,37]. Nowadays, among the APF, the only one preoperatively available is the DOI predicted by imaging [38]; in fact, combining different instrumental methodologies, such as magnetic resonance imaging and transoral ultrasound, can adequately estimate the measurement of tumor infiltration [39,40]. Thus, most of the surgical decision-making processes are based on this value. This measure can be considered a surrogate marker for tumor biology, as a more extensive deep infiltration reveals a more aggressive neoplasm, which is more prone to produce lymph node metastases than thick exophytic tumors [41]. Nevertheless, the DOI alone is insufficient to accurately predict the behavior of early lesions. Larson et al. [42] demonstrated how the simultaneous combination of different APFs, rather than the DOI value alone, affects DSS and OS in both univariate and multivariate analyses. Subramanian et al. [43] obtained comparable results, suggesting that even pT1N0 with free margins but presenting with simultaneous APFs should still be considered for adjuvant treatment. Furthermore, the reliability of DOI in predicting occult metastasis might vary widely according to the subsites involved [44]. In fact, among OTSCC patients, salient differences exist considering different subsites. In particular, those with OTSCC are more likely to experience cancer-specific mortality than patients with floor of the mouth, upper gum, and retromolar trigone cancer [45]. Several authors underlined how there are distinct anatomic barriers, local spreading pathways [46], and different nodal drainage patterns [47,48] that might justify this inferior survival outcomes for OTSCC. Accordingly, we can observe heterogeneity among the oral cavity subsites, even regarding APFs. Kim et al. [49] reported that PNI was more likely for tumors located in the oral tongue or floor of the mouth. Moreover, Liu et al. [50] demonstrated how the prognostic impact of pathological features was subsite-dependent, as PNI was associated with poor survival, especially in patients with OTSCC, where it was found to be the only significant factor associated with DSS of stage I and II. OTSCC’s tendency to present with PNI and its anatomical proximity with major cranial nerves might justify a more aggressive surgical approach, even in early small lesions [51]. PNI act as a passive conduit, providing an additional route for dissemination [52]. Therefore, PNI are among the worst prognosticators approved by the majority of investigators and are associated with locoregional recurrence, distant metastasis and decreased 5-year OS probability [53]. In addition, it has been demonstrated to be correlated with such major risk factors as extranodal extension [49]. All together, these findings may lead to planning a more aggressive therapeutic approach, even in early small OTSCCs where PNI is suspected [51]. Accordingly, the current National Comprehensive Cancer Network Panel (NCCN) Guidelines suggest the addition of adjuvant radiotherapy even for patients with surgically resected T1–T2 and N0 OTSCC when PNI is described in the pathology report [54]. 

In this scenario, observing the loss of tongue dorsum papillae in the area surrounding the tumor during the clinical examination has been emerging as an interesting predictor for APFs. To date in the literature, only Singh et al. investigated this topic, finding a significant relation between OTSCC with depapillation and pathologically documented PNI. Our study obtained similar results concerning depapillation correlation with PNI (*p* = 0.022). Observing this finding in a different population, heterogenic in terms of ethnicity and staging, reinforces the possible role of depapillation as a clinical predictive feature. This feature has the advantage of being cost effective, as it can be detected during a standard physical investigation. Particularly, despite the novelty of this feature, which physicians are not routinely trained to detect, in our study, the inter-rater agreement between the otolaryngologists was substantial, underlining how even less experienced specialists can identify it. Since our center is well-trained in applying NBI, an equivalent score has been obtained regarding Takano’s classification. Nevertheless, we did not observe any correlation between the NBI pattern and the depapillation group; therefore, we can state that the presence of depapillation does not hinder the effectiveness of NBI in detecting the typical perilesional vascular changes that occur during neoplastic development. Accordingly, also, no differences were found for what concerns surgical margins between the group of depapillated lesions and the group without atrophy of papillae. As the NBI evaluation was not affected by depapillation, we expected the same for margins, considering that in our center, we routinely tailor our resections based on NBI due to its established role in the whole oral cavity [20,55,56,57]. The automated delineation of tumor boundaries with this technique was recently initiated, and the effect of depapillation on that might be worth investigating [58]. 

Modern studies have demonstrated that PNI is a deliberate, molecularly mediated process that results from reciprocal interactions between cancer and nerves. Although depapillation is associated with PNI, it represents the mirror of the changes in the peritumoral microenvironment rather than an event driven purely by the progress of cancer alone. The first step of tumor infiltration and distant dissemination is a set of changes named epithelial-mesenchymal transition (EMT). EMT is a dynamic process allowing a polarized epithelial cell to undergo multiple biochemical changes leading to a mesenchymal cell phenotype, such as enhanced migratory capacity and invasiveness [59]; once the tumor infiltrates the basement membrane, it invades the adjacent cells first, which are the sustentacular cells. Since these cells clear the neurotransmitters synthesized by the taste receptor cells, there is a subsequent accumulation of neuronal promoters, leading to neurotropism and neural spread. This might explain why depapillation could be a clinical feature mirroring EMT, which leads to PNI and subsequent dissemination. 

Promising efforts have been made to anticipate APFs in a preoperative setting. Notwithstanding, unfavorable histologic parameters in preoperative biopsy specimens showed poor correlation with the subsequent resection specimen. Likewise, in early OSCC, the differentiation grade determined by biopsy was demonstrated to be of little predictive value for the grading of the resection specimen. Poor differentiation grade, revealed at the biopsy, could not be related to the presence of nodal metastasis or survival and seems not to have any prognostic value concerning outcome [60]. Regarding PNI and LVI, the preoperative biopsy specimens did not represent the final post-surgical specimen and the subsequent risk of occult metastasis [61]. Finally, the feasibility of predicting WPOI, a negative prognosticator of nodal metastases and oncological outcomes [31], in the preoperative biopsy specimen has been assessed by Pu et al. [31]. Despite the promising results, the authors did not obtain an adequate overlap between the intermediate biopsy pattern and the final WPOI of Type 4 and 5, whose clinical outcomes are relatively poor, making the biopsy pattern unreliable in predicting prognosis preoperatively. Although peritumoral depapillation is not ubiquitous when PNI is present, its assessment is time and cost-effective, and it could be easily integrated into standard clinical examination; furthermore, it is reproducible. The routine use of high-definition intraoperative endoscopy for NBI evaluation would allow the surgeon to better inspect the tumor’s surrounding areas and obtain magnified and more reliable images that allow an improved detection of depapillation. The retrospective nature of our study represents its main limitation together with the small study cohort. Lastly, NBI evaluation of the lesions is subjective, but we reduced this bias by letting different senior surgeons judge the appearance of neoplastic lesions on NBI filters.

## 5. Conclusions

In our study, clinical peritumoral depapillation is associated with PNI on the pathology report. Its presence does not affect the NBI’s ability to detect perilesional neoangiogenesis and delineate resection margins. In the future, combining different techniques and clinical features might allow us to depict an endoscopic signature of the more aggressive tumors and, eventually, tailor the surgical treatment. We hope that further studies will corroborate our findings and fully understand the process that leads to depapillation in roughly half of the OTSCC population.

## Figures and Tables

**Figure 1 cancers-15-01196-f001:**
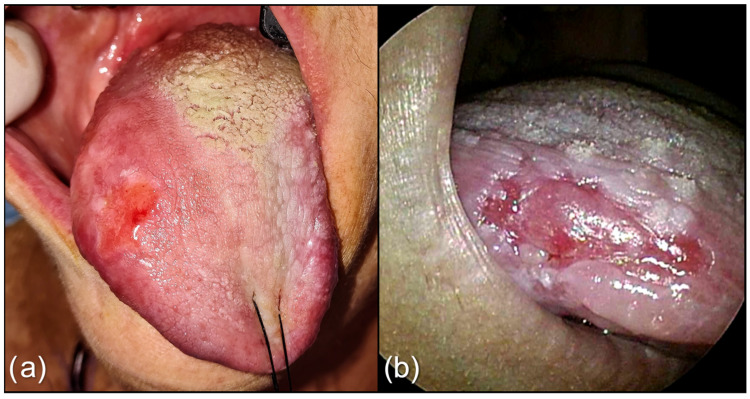
(**a**) Lesion of the right border and dorsum of tongue. An evident strip of depapillated mucosa surrounds the lesion; (**b**) Tongue cancer of the border of tongue on WL, with depapillated mucosa at the superior interface.

**Figure 2 cancers-15-01196-f002:**
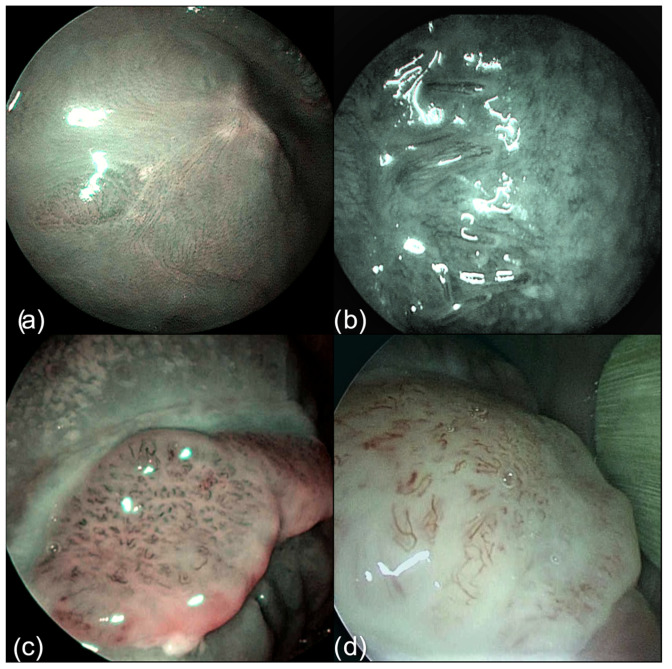
Takano’s classification of tongue vascular patterns on NBI filter. (**a**) Type I; (**b**) Type II; (**c**) Type III; (**d**) Type IV.

**Figure 3 cancers-15-01196-f003:**
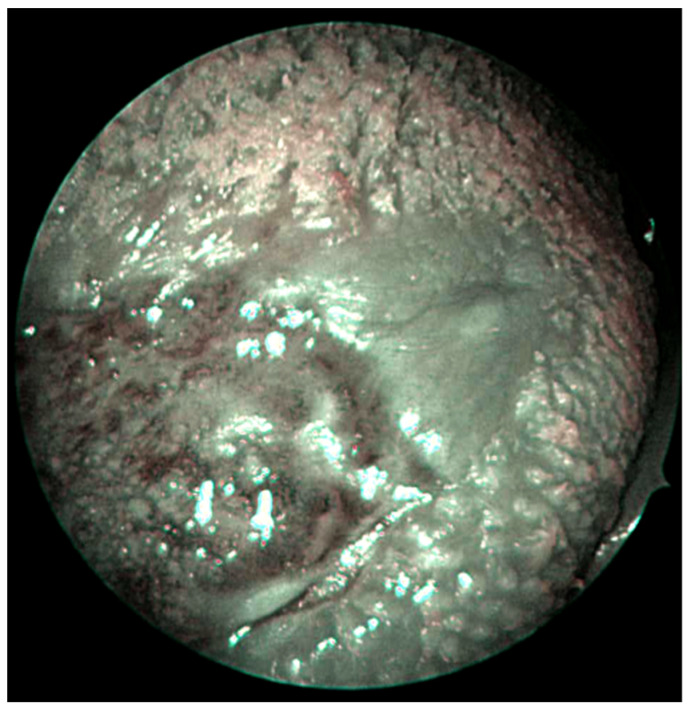
Dorsal tongue cancer on NBI filter (Takano’s classification Type III), with depapillated mucosa on the interface of the posterior margin.

**Figure 4 cancers-15-01196-f004:**
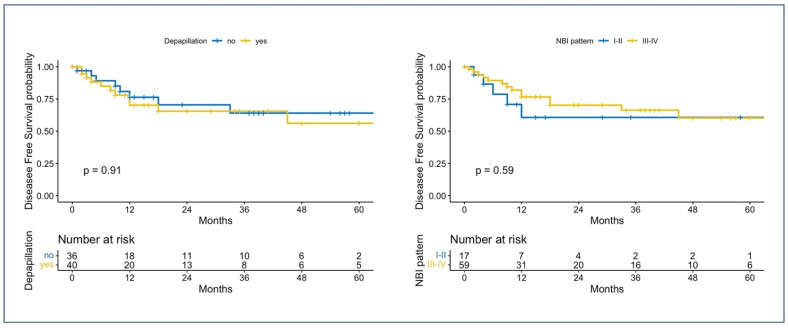
Kaplan–Meier survival curves for RFS. On the left, the RFS is compared based on the presence of depapillation; on the right, the RFS is compared according to the NBI vascular pattern. The *p*-values are computed using the log-rank tests.

**Table 1 cancers-15-01196-t001:** Patient demographics and tumor characteristics.

Characteristic		N = 76
Age	Median (IQR)	70.00 (56.00, 80.00)
	Range	22.00–100.00
Gender	F	40 (53%)
	M	36 (47%)
Smoking	No	44 (64%)
	yes	25 (36%)
Alcohol	No	54 (81%)
	yes	13 (19%)
cT	1	37 (49%)
	2	26 (34%)
	3	12 (16%)
	4a	1 (1.3%)
cN	0	68 (89%)
	1	1 (1.3%)
	2c	4 (5.3%)
	2b	2 (2.6%)
	3b	1 (1.3%)
Overall Stage	1	36 (47%)
	2	24 (32%)
	3	8 (11%)
	4a	6 (7.9%)
	4b	2 (2.6%)
pT	1	35 (47%)
	2	22 (29%)
	3	18 (24%)
pN	0	33 (75%)
	1	6 (14%)
	2b	2 (4.5%)
	2c	1 (2.3%)
	3b	2 (4.5%)
Margins	Free	37 (49%)
	Close	4 (5.3%)
	Positive	35 (46%)
Grading	1	13 (18%)
	2	51 (71%)
	3	8 (11%)
LVI *	No	55 (73%)
	Yes	20 (27%)
PNI *	No	44 (59%)
	Yes	31 (41%)
Pathological DOI *	Median (IQR)	6.00 (2.30, 10.00)
	Range	0.50–40.00
TILs *	No	19 (34%)
	Yes	37 (66%)
Budding	No	38 (62%)
	Yes	23 (38%)
WPOI *	1	1 (5.0%)
	2	2 (10%)
	3	8 (40%)
	4	7 (35%)
	5	2 (10%)
Depapillation	No	36 (47%)
	Yes	40 (53%)
NBI pattern	I–II	17 (22%)
	III–IV	59 (78%)
Aspect	Exophytic	30 (39%)
	Plane	23 (30%)
	Ulcerated	23 (30%)

* PNI = perineural invasion. LVI = lympho-vascular invasion. TILs = tumor-infiltrating lymphocytes. WPOI = worst pattern of invasion. DOI = depth of invasion.

**Table 2 cancers-15-01196-t002:** Univariable analysis according to the presence of depapillation.

Variable	Overall, N = 76	Depapillation		*p*-Value
		No, N = 36	Yes, N = 40	
cT				0.90
1	37 (49%)	19 (53%)	18 (45%)	
2	26 (34%)	12 (33%)	14 (35%)	
3	12 (16%)	5 (14%)	7 (18%)	
4a	1 (1.3%)	0 (0%)	1 (2.5%)	
cN				0.053
0	68 (89%)	33 (92%)	35 (88%)	
1	1 (1.3%)	0 (0%)	1 (2.5%)	
2b	4 (5.3%)	0 (0%)	4 (10%)	
2c	2 (2.6%)	2 (5.6%)	0 (0%)	
3b	1 (1.3%)	1 (2.8%)	0 (0%)	
Clinical stage				0.95
1	36 (47%)	18 (50%)	18 (45%)	
2	24 (32%)	11 (31%)	13 (32%)	
3	8 (11%)	4 (11%)	4 (10%)	
4a	6 (7.9%)	2 (5.6%)	4 (10%)	
4b	2 (2.6%)	1 (2.8%)	1 (2.5%)	
pT				0.94
1	35 (47%)	17 (47%)	18 (46%)	
2	22 (29%)	11 (31%)	11 (28%)	
3	18 (24%)	8 (22%)	10 (26%)	
pN				0.91
0	33 (75%)	14 (82%)	19 (70%)	
1	6 (14%)	2 (12%)	4 (15%)	
2b	2 (4.5%)	0 (0%)	2 (7.4%)	
2c	1 (2.3%)	0 (0%)	1 (3.7%)	
3b	2 (4.5%)	1 (5.9%)	1 (3.7%)	
Pathological stage				0.59
1	32 (43%)	15 (43%)	17 (44%)	
2	18 (24%)	10 (29%)	8 (21%)	
3	19 (26%)	9 (26%)	10 (26%)	
4a	3 (4.1%)	0 (0%)	3 (7.7%)	
4b	2 (2.7%)	1 (2.9%)	1 (2.6%)	
Margins				0.49
free	37 (49%)	15 (42%)	22 (55%)	
close	4 (5.3%)	2 (5.6%)	2 (5.0%)	
positive	35 (46%)	19 (53%)	16 (40%)	
Grading				0.49
1	13 (18%)	8 (24%)	5 (13%)	
2	51 (71%)	22 (65%)	29 (76%)	
3	8 (11%)	4 (12%)	4 (11%)	
LVI *				0.060
no	55 (73%)	30 (83%)	25 (64%)	
yes	20 (27%)	6 (17%)	14 (36%)	
PNI *				0.022
no	44 (59%)	26 (72%)	18 (46%)	
yes	31 (41%)	10 (28%)	21 (54%)	
Pathological DOI *				0.73
Median (IQR)	6.00 (2.30, 10.00)	5.50 (2.25, 9.75)	6.00 (3.00, 10.50)	
TILs *				0.64
no	19 (34%)	10 (37%)	9 (31%)	
yes	37 (66%)	17 (63%)	20 (69%)	
Budding activity				0.67
no	38 (62%)	17 (65%)	21 (60%)	
yes	23 (38%)	9 (35%)	14 (40%)	
WPOI *				0.25
1	1 (5.0%)	0 (0%)	1 (12%)	
2	2 (10%)	2 (17%)	0 (0%)	
3	8 (40%)	3 (25%)	5 (62%)	
4	7 (35%)	5 (42%)	2 (25%)	
5	2 (10%)	2 (17%)	0 (0%)	
NBI pattern				0.56
I–II	17 (22%)	7 (19%)	10 (25%)	
III–IV	59 (78%)	29 (81%)	30 (75%)	
Aspect				>0.99
exophytic	30 (39%)	14 (39%)	16 (40%)	
plane	23 (30%)	11 (31%)	12 (30%)	
ulcerated	23 (30%)	11 (31%)	12 (30%)	

* PNI = perineural invasion. LVI = lympho-vascular invasion. TILs = tumor-infiltrating lymphocytes. WPOI = worst pattern of invasion. DOI = depth of invasion.

## Data Availability

The data can be shared up on request.

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
