# Peer review of "The Role of Peritumoral Depapillation and Its Impact on Narrow-Band Imaging in Oral Tongue Squamous Cell Carcinoma"

_cancers, 2023, doi:10.3390/cancers15041196_

Round 1

Reviewer 1 Report (Previous Reviewer 4)

Dear authors, despite I appreciate your modifications, you did not address my concerns in the previous version.  The rationale of your paper is based only a questionable paper (ref. 17 in the text). The addition of another subjective variable (NBI patterns) further weakens the evidence behind any incidentally found association.

Reviewer 2 Report (Previous Reviewer 3)

This study examined the correlation between tongue depapillation and nerve infiltration. The number of cases is small and the lesions are varied, so further studies are needed, but the important results should be published.

Reviewer 3 Report (Previous Reviewer 2)

Dear authors, all points have been answered adequately.

Reviewer 4 Report (Previous Reviewer 1)

The author has provided relevant references to fill in the missing links in the previous manuscript. 

This manuscript is a resubmission of an earlier submission. The following is a list of the peer review reports and author responses from that submission.

Round 1

Reviewer 1 Report

Andrea Iandelli and colleagues present the association between depapillation and perineural invasion (PNI), an independent prognostic factor of oral tongue squamous cell carcinoma (OTSCC). In addition, the authors also showed that the narrow band imaging (NBI) performance is not affected by depapillation. This manuscript provided valuable information that potentially guides and improves the OTSCC treatment, especially dispelling the concerns that depapillation may hinder NBI accuracy. Overall, the work is straightforward and easy to follow; however, there are some significant weaknesses of this study that would have to be addressed before this is acceptable for publication.

1)       Author raised the concern that depapillation would potentially change epithelium thickness which may affect the NBI performance, while this statement is not well supported by any reference or data provided. Owing to this statement being related to the hypothesis to be proven in this work, more information has to be provided to consolidate this statement.

2)       Another important point to consider is that in the result section, the author showed that there is not any correlation between the NBI pattern and the depapillation group. While in the conclusion section, the author claim “Its (depapillation) presence does not affect the NBI ability to detect perilesional neoangiogenesis…… .”. The result does not well support the above statement (the term “perilesional neoangiogenesis” even did not mention in any other place in the manuscript. Unless the author could provide additional information that supports the NBI pattern is solely determined by perilesional neoangiogenesis, otherwise, the statement in the conclusion section should be toned down.

Other minor aspects:

On lines 64-66, “It is believed that the local neoplastic invasion and inflammation would cause the tongue’s papilla dysfunction, leading to the clinical appearance of depapillation [18]”. The clinical appearance of depapillation is reported in reference 17. Reference 18 mainly focuses on the inflammation event. Please double-check if reference 17 should be quoted in this statement.

On line 147, the IPCL is already defined on line 73. Please use the abbreviations on line 147.

On lines 201-202, “When grouping the patients according to the presence of depapillation, the latter was strongly correlated to PNI, …”. It is unclear which group that “latter” is referring to.

On line 209, for consistency, RFS should be used instead of recurrence-free survival.                           

On line 213, First row of Table 1, “N = 76 1” please delete the “1”.  

Reviewer 2 Report

Dear authors, a well designed, properly presented retrospective study on possible correlation of depapillation of the tongue peripheral to diagnosed SCC. Minor flaws can be detected according to STROBE guidelines of presenting a cohort study. Overall, a very well manuscript publishable worthy.

Reviewer 3 Report

This article examines the relationship between tongue depapillation and perineural invasion. It also examines the relationship between NBI and depapillation. It is clinically interesting.

Intoroduction. 

Throughout, the concept of tongue depapillation is a little unclear. Is it a non-homogeneous lesion? Please make it clear definition.

Material & Methods 

The classification of Tanakno et al. is used, but this classification is difficult when the mucosa is thick and the blood vessels may not be seen at all in the first place. See e.g. (Ota et al. 2022 cancers).

Results

Many of the cases have a positive surgical margin, could this be related to PNI?

What are the findings of biological staining, such as iodine staining, if any?

Reviewer 4 Report

You have added another variable (retrospectively extracted from your records) and tried to correlate it with other clinical-pathological factors. While I commend your efforts, I don't think the analysis offers anything new to the readers. The finding of depapillation is quite common also in benign conditions such as geographic tongue, and many dermatological diseases (psoriasis etc.). It is also associated with the food hypersensitivity. Since you cannot control for these confounders and that the relationship of depapillation with oral cancer is very preliminary, your mix between these findings and the NBI patterns brings only more confusion in this field.